# ARITHMETIC-BENCH: EVALUATING MULTI-STEP REASONING IN LLMS WITH BASIC ARITHMETIC

## ABSTRACT

We propose Arithmetic-Bench, a benchmark designed to evaluate the multi-step reasoning ability of large language models (LLMs) through basic arithmetic operations. The benchmark covers fundamental mathematical operations such as addition, subtraction, multiplication, and division, while also incorporating subtasks like copying, reversing, counting, and base conversion. Experimental results show that the accuracy of current LLMs drops sharply when performing arithmetic operations involving more than 10 digits, implying a failure of generalization in multi-step reasoning. We further analyze the root causes of these failures. While LLMs can achieve a certain degree of arithmetic generalization through training on limited-length sequences, they fail to generalize to arbitrary lengths. This is due to the inherent complexity of arithmetic tasks: achieving true arithmetic generalization cannot rely on memorization alone but requires the acquisition of genuine reasoning mechanisms. Compared to other math benchmarks, Arithmetic-Bench provides a simple and fair framework. Because the tasks are purely synthetic, they are easy to generate and largely free from human biases. We believe that arithmetic tasks are both fundamental and necessary for advancing reasoning models, and Arithmetic-Bench offers a principled way to evaluate them.

## 1 INTRODUCTION

The rapid development of large language models (LLMs) has led to significant progress in natural language understanding and generation. However, despite their strong performance on existing reasoning benchmarks such as AIME Veeraboina (2023), GSM8K Cobbe et al. (2021), and MATH Hendrycks et al. (2021), these models often struggle with basic arithmetic tasks. This inconsistency raises critical questions about the nature of reasoning in LLMs: do they truly possess multi-step reasoning capabilities, or are they merely performing pattern matching based on training data?

### 1.1 DISADVANTAGES OF MATH BENCHMARKS

There are a lot of existing math-related datasets and benchmarks. However, in practical applications involving reasoning models, we have observed several limitations in these math benchmarks.

**Hard to Collect.** Although a large number of math problems are available online, their difficulty levels and coverage are difficult to control precisely, often resulting in datasets with uneven quality and distribution. Manually creating problems is costly and labor-intensive, while the reliability of model-generated problems remains uncertain. In addition, manually collected problems are inevitably subject to human biases.

**Hard to Decontaminate (Easy to Cheat).** Given the vast amount of data on the internet, it is almost inevitable that identical or highly similar problems already exist. Furthermore, it is difficult to prevent individuals or organizations from intentionally training models on benchmark data to inflate performance.

**Hard to Evaluate.** Evaluation poses significant challenges. For open-ended problems, such as proofs, relying on models for evaluation is unreliable and vulnerable to hacking. For problems with definitive answers, such as computational tasks, formatting problems necessitate complex pattern-matching methods to verify correctness, which are error-prone and cause fluctuations in evaluation results. This makes it difficult to determine whether a model's reasoning ability has truly improved.

**Hard to Scale.** Constructing a smooth difficulty progression is highly challenging. Some problems are difficult due to reliance on obscure knowledge, while others require multi-step reasoning. Consequently, certain problems primarily test memory rather than reasoning. Since these two types of difficulty differ in nature, they cannot be directly compared or quantified.

### 1.2 ADVANTAGES OF ARITHMETIC BENCHMARKS

In contrast to complex mathematical operations, arithmetic operations serve as a natural testbed for reasoning because they are deterministic, require structured multi-step execution, and have clear correctness criteria.

**Easy to Collect.** Generating arithmetic expressions is very straightforward, and results obtained through a calculator are guaranteed to be correct, ensuring the quality of the problems. More importantly, these problems are purely synthetic, which greatly reduces human bias.

**Easy to Decontaminate (Hard to Cheat).** Thanks to the nature of large numbers, no special filtering is required. Memorizing answers provides no advantage in large-number arithmetic: even if one memorizes all two-digit multiplications, they would cover only 1% of three-digit multiplications. Moreover, brute-force memorization inevitably leads to forgetting.

**Easy to Evaluate.** There is no ambiguity: evaluation can be performed directly with a simple check (e.g., `a in b`), without additional prompts, since all mainstream models already know basic arithmetic and the numbers do not suffer from formatting issues.

**Easy to Scale.** Arithmetic tasks can be scaled to arbitrary digit lengths and varying levels of complexity, which creates a continuous difficulty curve. This makes it possible to evaluate a model's true reasoning ability, going beyond mere memorization.

### 1.3 MEMORIZATION VS GENERALIZATION

We further raise the following two key questions and address them using Arithmetic-Bench.

**How can we verify whether the improvements of existing LLMs on reasoning benchmarks may come from memorizing the answers?**

It is possible to train a model on a finite math benchmark dataset and then achieve very high scores on that benchmark. However, even if we train on a multiplication benchmark dataset, the model still cannot achieve high scores on the multiplication benchmark, because the multiplication benchmark is randomly generated from a space that far exceeds the model's capacity limit.

**How can we construct tasks that cannot be solved by memorizing the answers?**

The information required to fully memorize long multiplication numbers is infinite, whereas the information needed to memorize the rules of multiplication is finite. The space of multiplication is so large that, unlike Olympiad math problems which are finite, it cannot be completely memorized; only by understanding the rules of multiplication can true generalization be achieved.

### 1.4 PROXY METRIC

In the field of image generation, text rendering seems like a minor skill, but the Nano Banana Google (2025) team treated it as an important metric. Text is highly structured, and even small stroke errors are obvious, making it a strict test of precision. Mastering text forces the model to control structure and detail at the pixel level, which then improves general quality. By using text rendering as a proxy metric, the team showed how optimizing for a highly demanding, low-tolerance subtask can push models to develop transferable skills that enhance broader performance.

Arithmetic-Bench is also this type of task, requiring models to have stable and precise reasoning abilities, which are necessary for solving truly complex problems, such as Fermat's Last Theorem. The formal proof of Fermat's Last Theorem contains tens of thousands of lines of Lean code Buzzard & contributors (2025); de Moura & Ullrich (2021), and even without considering the details of each step, it still represents an extremely long chain of reasoning. Therefore, we believe that Arithmetic-Bench is suitable as a proxy metric for mathematical reasoning.

Whether a model truly has reasoning ability is a rather vague question, but if it can handle large-number arithmetic well, it can then be considered to possess a certain level of reasoning ability.

The contributions of this paper are as follows:

- We introduce **Arithmetic-Bench**, a benchmark consisting of basic arithmetic tasks designed to evaluate models' multi-step reasoning and computational skills.

- We provide a theoretical analysis of the connection between arithmetic tasks and reasoning ability, and empirically demonstrate their correlation.

- We benchmark multiple mainstream models, showing that current models perform poorly on these tasks, underscoring the need to improve multi-step reasoning capabilities.

## 2 RELATED WORK

### 2.1 MATH BENCHMARK

AIME Veeraboina (2023) (American Invitational Mathematics Examination) consists of competition-level math problems, covering advanced algebra, number theory, combinatorics, and geometry. It includes 15 problems per year before 2000 and 30 problems per year thereafter.

MATH Hendrycks et al. (2021) contains high-school level math problems spanning algebra, calculus, number theory, and more, totaling 12,500 problems. Large language models (LLMs) still struggle on some of these problems, particularly those requiring multi-step reasoning, achieving only moderate accuracy.

CMATH Wei et al. (2023) is a dataset of Chinese elementary school math word problems, comprising 1.7k problems with detailed annotations sourced from real workbooks and exams.

GSM8K Cobbe et al. (2021) (Grade School Math 8K) is a set of 8,000 elementary-level math problems. Current LLMs can perform well on this benchmark, achieving over 97% accuracy through prompt engineering Zhong et al. (2024).

GSM-Symbolic Mirzadeh et al. (2024) is a variant of GSM8K in which numbers are replaced with random values. The resulting performance drop indicates that models may rely on memorized numbers and patterns.

These benchmarks cover a broad spectrum from elementary arithmetic to advanced competition-level mathematics.

### 2.2 ARITHMETIC BENCHMARK

Benchmarks focusing specifically on arithmetic, such as Math401 Yuan et al. (2023) and the arithmetic subset of BIG-Bench Srivastava et al. (2023), evaluate basic operations but have two key limitations: (i) potential memorization due to fixed datasets, and (ii) limited length generalization, since most problems involve numbers with fewer than ten digits. To address these issues, some approaches use synthetic math games Kurtic et al. (2024), though these often require complex rules and careful prompt design. In contrast, Arithmetic-Bench provides a simpler framework for evaluating arithmetic reasoning with controlled difficulty and sequence length.

### 2.3 ARITHMETIC REASONING BASED ON DEEP LEARNING

Early works, including Neural GPU Łukasz Kaiser & Sutskever (2016) and Neural Turing Machine Graves et al. (2014), improved algorithm execution by designing specialized architectures such as recursive convolutional networks and memory modules. More recent methods, such as Goat Liu & Low (2023) and MathGLM Yang et al. (2023), train LLMs on carefully constructed arithmetic datasets, while other approaches, like Scratchpad Nye et al. (2021), leverage techniques such as chain-of-thought (CoT) reasoning Wei et al. (2022) and curriculum learning Bengio et al. (2009). These methods enhance arithmetic performance for numbers within certain digit lengths, but they generally fail to generalize to longer sequences, highlighting the challenge of length extrapolation in arithmetic reasoning.

## 3 ARITHMETIC-BENCH

### 3.1 CAPACITY

Mathematical reasoning is based on axioms and deductive rules. Here, axioms provide fundamental assumptions, and deductive rules specify how to derive new facts from existing ones. A proposition is considered correct if it can be derived from the axioms. The key difference between reasoning and common sense lies in the number of inference steps: common sense typically requires only a single step, whereas reasoning involves multiple iterative steps. Following Zhou (2025), reasoning models are characterized by producing intermediate reasoning tokens before generating the final output.

From these observations, we propose the following definitions:

**Definition 1.**  Reasoning is the iterative application of operations on finite information, where each operation transforms known information into new information.

**Definition 2.**  Arithmetic is a special case of reasoning, where the operations are derived from a finite lookup table of number operations.

Clearly, arithmetic over natural numbers satisfies Definition 1 and is therefore a form of reasoning.

A task with finite information can be fully learned by memorizing all cases, provided the model has sufficient capacity. Here, capacity refers to the total amount of information that a model can store or represent in its parameters. More concretely, if a model has $N$ parameters and each parameter can store approximately $c$ bits of information independently, then the model capacity is roughly $C = N \cdot c$ bits. When the information content of a task exceeds the model's capacity, the task becomes unlearnable due to inevitable forgetting. This is formalized by the following principle:

**Theorem 1.**  A container with capacity $a$ cannot hold information exceeding $a$.

For example, a model with 400 parameters can store the $9 \times 9$ multiplication table; a model with 20,000 parameters can fully memorize the first 10,000 digits of $\pi$. In contrast, a model with 10,000 parameters can memorize only about 70% of these digits, regardless of training duration. This is coincidence with the fact that current language models can and only can store 2 bits of knowledge per parameter Allen-Zhu & Li (2024).

Next, we relate arithmetic performance to general reasoning ability.

**Theorem 2.**  If a model cannot learn an arithmetic problem, it cannot learn a reasoning problem of equivalent complexity.

**Proof.**  Any reasoning task can be encoded as an equivalent arithmetic problem by mapping basic operations to numbers. If a model can solve this arithmetic problem, it can solve the corresponding reasoning task. By Theorem 1, if the model cannot solve the arithmetic problem, it lacks sufficient capacity to represent the necessary information, and therefore cannot learn any reasoning problem of equal or greater complexity.

Computational stability can be analyzed similarly. Suppose each operation introduces a small error $\epsilon$, and the task can tolerate an expected error $\delta$. Then, only a limited number of operations can be performed before the accumulated error exceeds $\delta$, defining the model's computational capacity.

A model can reliably complete a reasoning task only if both its information capacity and computational capacity are sufficient. Notably, increasing the number of digits in arithmetic primarily challenges computational capacity rather than information capacity. Therefore, benchmarks like Arithmetic-Bench can probe reasoning ability beyond the limits of information storage by evaluating tasks that require extensive computation.

### 3.2 ERROR ACCUMULATION

Assuming the model has sufficient information capacity and fully understands the reasoning rules, errors may still occur due to probabilistic predictions. To mitigate accumulated errors during iterative reasoning, verification strategies can be employed. Let the probability of making a mistake in a

single computation be $p$, assuming independent computations. Without verification, the probability of submitting an incorrect result is

$$P_{\text{error, no verification}} = p.$$

If one additional independent verification is performed, an undetected error occurs only if the first computation is wrong, the verification is also wrong, and the two errors coincide exactly. Let $q$ denote the conditional probability that two independent errors yield the same incorrect result ($0 < q < 1$). Then, the probability of an undetected error under verification is

$$P_{\text{error, verification}} = p^2 q.$$

Since $0 < p < 1$ and $0 < q < 1$, it follows that

$$P_{\text{error, verification}} = p^2 q < p = P_{\text{error, no verification}}.$$

Therefore, verification reduces the probability of undetected errors.

For instance, if $p = 0.1$ and $q = 0.1$, the error probability without verification is $10\%$, while with verification it decreases to $0.1\%$, representing a reduction by two orders of magnitude. This illustrates that, for reasoning tasks, implementing verification can be more effective than merely increasing the number of reasoning steps or output tokens.

### 3.3 DESIGN

Arithmetic-Bench is a dynamic benchmark that generates arithmetic problems of varying lengths and complexities. It includes binary and unary operations as shown in Table 1 and Table 2:

Each problem is randomly generated to ensure that tasks cannot be memorized, and all require multi-step iterative operations. The benchmark only requires the model to have basic mathematical knowledge, without relying on any advanced theorems to eliminate the influence of memorized knowledge on problem difficulty. Prompts are kept as simple as possible to minimize the influence of prompt- or instruction-following abilities on the results. Evaluation is performed directly using 'a in b', which is simple and accurate. Different models may format their outputs differently; for example, DeepSeek outputs answers in '\boxed{}' and GPT prefers bold answers using '**', but 'a in b' can match any similar format. If a model produces the correct intermediate result during the process, it indicates that the model's reasoning can reach the final answer. Since the probability of guessing large-number results correctly in the middle of the process is extremely low, this does not compromise fairness. Some models, such as DeepSeek, may output answers separated by symbols like '",//!".' We remove all such symbols from the results before matching the answers. The steps for decimal addition and multiplication are basically the same as for integers, differing only in the decimal point shift. Therefore, only integer operations are considered. To ensure comparable computational complexity between division and multiplication and to avoid results that are too small, we perform division of $2n$-digit numbers by $n$-digit numbers. Modular operations (`mod`) and division steps are essentially the same, so only division is considered. Exponentiation (`pow`) is too computationally expensive and is therefore not considered. We primarily evaluate the model's arithmetic performance from the following two dimensions:

**Accuracy**
Full-match accuracy is used, without considering digit-wise accuracy, since in large-number operations models often produce outputs with incorrect digit lengths, making alignment with the correct answer difficult.

**Length Generalization Curve**
This curve illustrates the relationship between model accuracy and the number of digits in the input. It provides insight into how well the model can generalize to longer sequences, indicating its computational capacity.

### 3.4 PROMPT

The prompts for different tasks are shown in Table 3. They are designed to be as simple as possible to enhance readability for both humans and models.

Table 1: Main tasks: standard arithmetic tasks.

| Task | Description |
|---|---|
| Add, Sub, Mul, Div | Integer addition, subtraction, multiplication, and division of 2 numbers, where both operands are $n$-digit integers. These tasks evaluate the model's ability to perform standard arithmetic operations and its accuracy and consistency across multiple digits. |
| Add_1, Sub_1, Mul_1, Div_1 | Integer addition, subtraction, multiplication, and division of 2 numbers, where one operand is an $n$-digit integer and the other is a single-digit integer. Since $n \times n$ multiplication can be decomposed into multiple $n \times 1$ multiplications, this task is used for evaluation. |

Table 2: Sub tasks: basic operations related to arithmetic.

| Task | Description |
|---|---|
| Copy | In multi-step reasoning, the model often needs to repeat operands multiple times. This task evaluates the model's ability to correctly copy values within a reasoning chain. |
| Rev, Space | Data representation significantly affects performance Lee et al. (2023). For instance, columnar (vertical) arithmetic is written from right to left, which can hinder next-token prediction. Reversal and splitting operations, such as little-endian storage or separating numbers into individual characters, are evaluated to test the model's adaptability to different representations. |
| Count, Len | Models sometimes produce outputs of incorrect length. These tasks test the model's counting ability. Since the answers are small numbers, parentheses are used to prevent models from accidentally guessing the correct answer during generation. |
| Box | Some operations, like `count` and `len`, require formatted output. This task evaluates the model's ability to correctly insert parentheses as a formatting symbols. |
| B2D, D2B | Neural GPU Łukasz Kaiser & Sutskever (2016) has shown better performance in binary than decimal. These tasks evaluate the model's ability to convert between binary and decimal representations. |

## 3.5 GENERATION

The generation of Arithmetic-Bench is very straightforward: two numbers are randomly generated and then concatenated using prompt templates for different tasks. The pseudocode is as follows.

---
**Algorithm 1** Generate Arithmetic Dataset (gen_2)
---

1: **procedure** GEN_2(fun, n, d)
2:     **for** $digits \leftarrow 1$ **to** $d$ **do**
3:         **for** $i \leftarrow 1$ **to** $n$ **do**
4:             **if** fun = div **then**
5:                 $a \sim \text{Uniform}(10^{2 \cdot digits - 1}, 10^{2 \cdot digits} - 1)$
6:                 $b \sim \text{Uniform}(10^{digits - 1}, 10^{digits} - 1)$
7:             **else**
8:                 $a \sim \text{Uniform}(10^{digits - 1}, 10^{digits} - 1)$
9:                 $b \sim \text{Uniform}(10^{digits - 1}, 10^{digits} - 1)$
10:             **end if**
11:             $c \leftarrow fun(a, b)$
12:             Output sample $(digits, a, b, c)$
13:         **end for**
14:     **end for**
15: **end procedure**

---

Table 3: Arithmetic Prompts

| Task | Prompt |
|------|--------|
| Add | $a + b =?$ |
| Sub | $a - b =?$ |
| Mul | $a * b =?$ |
| Div | Perform integer division: $a/b =?$ |
| Copy | Copy the following number: $a$ |
| Rev | Reverse the following number: $a$ |
| Box | Put the following number in parentheses only: $a$, example: (number) |
| Space | Insert a space between every digit in the following number: $a$ |
| Len | How many digits are in the following number: $a$, put answer in parentheses only, example: (number) |
| Count | How many 0 are in the following number: $a$, put answer in parentheses only, example: (number) |
| B2d | Convert the following binary number to decimal: $a$ |
| D2b | Convert the following decimal number to binary: $a$ |

# 4 EXPERIMENTAL RESULTS

## 4.1 SETUP

We compared several state-of-the-art models:

LLaMA series Dubey et al. (2024), Qwen series Yang et al. (2024; 2025); Team (2025), DeepSeek series Guo et al. (2025), GPT series OpenAI (2023); Hurst et al. (2024)

Both open-source and closed-source models were used to ensure a comprehensive evaluation.

All tasks were tested with problems randomly generated for each digit length from 1 to 100. For Qwen and LLaMA, $n = 10$ problems were generated per digit length. For DeepSeek and GPT, due to resource limitations and slower inference speed, $n = 1$ problem per digit length was used.

Table 4: Model performance on Arithmetic-Bench (Main tasks)

| Model | add | sub | mul | div | add_1 | sub_1 | mul_1 | div_1 |
|-------|-----|-----|-----|-----|-------|-------|-------|-------|
| Llama-3-8B-Instruct | 11.7% | 10.0% | 1.9% | 1.8% | 93.4% | 93.5% | 20.3% | 18.4% |
| Llama-3-70B-Instruct | 20.5% | 16.3% | 2.2% | 2.2% | 93.4% | 91.9% | 27.2% | 26.4% |
| Qwen2.5-0.5B-Instruct | 8.1% | 7.3% | 1.6% | 1.4% | 23.1% | 19.1% | 17.0% | 18.1% |
| Qwen2.5-1.5B-Instruct | 12.6% | 11.6% | 2.0% | 1.7% | 69.4% | 67.8% | 27.3% | 21.5% |
| Qwen2.5-3B-Instruct | 11.7% | 14.1% | 1.9% | 1.7% | 77.6% | 71.1% | 30.7% | 31.5% |
| Qwen2.5-7B-Instruct | 25.7% | 20.3% | 2.2% | 2.9% | 89.2% | 86.5% | 45.2% | 45.2% |
| Qwen2.5-14B-Instruct | 28.9% | 32.3% | 2.3% | 3.2% | 98.0% | 97.7% | 66.2% | 77.9% |
| Qwen2.5-32B-Instruct | 50.2% | 31.3% | 2.5% | 4.2% | 96.9% | 97.5% | 79.1% | 79.0% |
| Qwen2.5-72B-Instruct | 31.5% | 30.8% | 2.5% | 4.3% | 98.0% | 96.9% | 51.5% | 48.5% |
| DeepSeek-R1-Distill-Llama-8B | 9.0% | 8.0% | 3.0% | 2.0% | 75.0% | 65.0% | 19.0% | 18.0% |
| DeepSeek-R1-Distill-Llama-70B | 14.0% | 14.0% | 3.0% | 5.0% | 93.0% | 91.0% | 27.0% | 25.0% |
| DeepSeek-R1-Distill-Qwen-1.5B | 10.0% | 12.0% | 4.0% | 3.0% | 44.0% | 64.0% | 23.0% | 21.0% |
| DeepSeek-R1-Distill-Qwen-7B | 14.0% | 11.0% | 4.0% | 3.0% | 77.0% | 77.0% | 32.0% | 32.0% |
| DeepSeek-R1-Distill-Qwen-14B | 13.0% | 18.0% | 4.0% | 4.0% | 78.0% | 77.0% | 27.0% | 23.0% |
| DeepSeek-R1-Distill-Qwen-32B | 21.0% | 26.0% | 4.0% | 7.0% | 83.0% | 75.0% | 42.0% | 43.0% |
| DeepSeek-R1-671B | 46.0% | 58.0% | 10.0% | 10.0% | 100.0% | 99.0% | 56.0% | 69.0% |
| QwQ-32B | 26.0% | 26.0% | 11.0% | 10.0% | 99.0% | 96.0% | 41.0% | 69.0% |
| Qwen3-235B-A22B | 41.0% | 40.0% | 10.0% | 11.0% | 100.0% | 100.0% | 58.0% | 78.0% |
| gpt-4 | 51.0% | 38.0% | 3.0% | 4.0% | 100.0% | 99.0% | 61.0% | 74.0% |
| gpt-4o | 68.0% | 84.0% | 3.0% | 3.0% | 100.0% | 100.0% | 85.0% | 79.0% |
| gpt-3.5 | 15.0% | 21.0% | 3.0% | 3.0% | 97.0% | 89.0% | 29.0% | 48.0% |

## 4.2 ANALYSIS

Main results are shown in Table 4. The accuracy of addition and subtraction is comparable, as is that of multiplication and division. Multiplication is significantly more challenging than addition, with accuracy roughly proportional to the maximum number of digits the model can handle. On multiplication tasks, the best models, Deepseek-R1, QwQ and Qwen3, can correctly solve numbers with up to 10 digits.

Tasks involving $n \times 1$-digit numbers are relatively easier, yet most models still fail to achieve 100% accuracy. Tasks where models perform relatively well include **add_1, sub_1, copy, box, and space**, with some models reaching perfect accuracy. These tasks share a common feature: they do not require complex reasoning, and the input-output structures are largely similar. For example, in **add_1** and **sub_1**, changes mostly occur in the last digits.

**Limitations of Reasoning.** Reasoning models, such as Deepseek-R1, QwQ and Qwen3, outperform non-reasoning models on tasks like multiplication and base conversion, but underperform on simpler tasks, such as addition and single-digit multiplication. This phenomenon is consistent with the observations reported in Shojaee et al. (2025): on low-complexity tasks, non-reasoning models outperform reasoning models; on medium-complexity tasks, reasoning models demonstrate an advantage; and on high-complexity tasks, both types of models experience complete failure.

**Influence of Scaling.** Within the same model series, larger models generally perform better on arithmetic tasks. However, Qwen-72B does not outperform Qwen-32B, suggesting that merely increasing model size does not necessarily resolve arithmetic challenges.

**Influence of Distillation.** The DeepSeek distilled models perform worse than their corresponding Qwen counterparts on simple arithmetic tasks, like addition and subtraction, and only marginally outperform Qwen on multiplication and counting. This indicates limitations in their reasoning ability, suggesting that the full reasoning capability of a large model may not have been successfully distilled into these smaller models.

GPT-4's average performance falls between Qwen2.5 and Qwen3, indicating that closed-source models do not necessarily demonstrate stronger arithmetic capabilities. Overall, the accuracy of all models remains relatively low, and true generalization in arithmetic has yet to be achieved.

## 4.3 LENGTH GENERALIZATION

As shown in Figure 1, accuracy decreases significantly as the number of digits increases. For multiplication, once the number of digits exceeds a certain threshold, models consistently fail to produce correct results. Therefore, the overall accuracy is approximately equal to the maximum number of digits in multiplication that the model can handle.

Even the largest and most advanced models, including Qwen3, DeepSeek, and GPT-4, continue to struggle with arithmetic tasks at scale. Specifically, they are unable to reliably solve 10-digit multiplication and often fail at 100-digit addition, despite their strong performance on a wide range of natural language tasks. This indicates that scaling alone does not resolve the fundamental challenges of arithmetic reasoning, and that current architectures still lack robust mechanisms for exact, length-generalizable computation.

## 4.4 MEMORIZATION OF FINITE DATASETS

As shown in Figure 2, training on AIME test set can push accuracy to 100%. We also observed similar phenomena on other finite datasets, demonstrating that finite benchmarks are prone to cheating. Notably, it requires around 100 epochs to memorize well, rather than remembering it after a single pass. Even after reaching 100% accuracy, fluctuations may still occur.

## 4.5 CORRELATION BETWEEN REASONING AND ARITHMETIC

As shown in Figure 5, The model's performance on mathematical benchmarks such as AIME is positively correlated with its performance on large-number multiplication. Reasoning models ex-

hibit stronger multiplication ability compared to non-reasoning models, but perform worse than non-reasoning models on simple tasks such as addition.

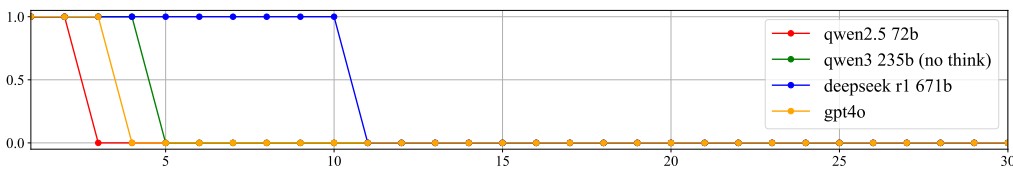

Figure 1: Length Generalization Curve of Multiplication, x axis is length from 1 to 30, y axis is accuracy.

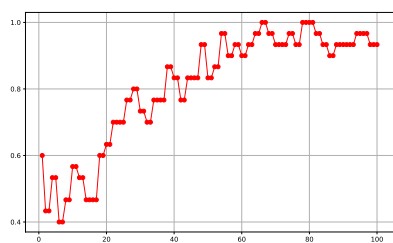

Figure 2: Results of training on AIME 2024, x axis is epoch, y asix is accuracy.

Table 5: Comparison of performance on Multiplication and AIME 2024

| Model | Mul Acc (%) | AIME Acc (%) |
|---|---|---|
| Qwen2.5 72b | 2 | 13.5 |
| Qwen3 235b (no think) | 4 | 40.1 |
| Qwen3 235b (think) | 10 | 85.7 |
| QwQ | 11 | 79.5 |
| Deepseek r1 671b | 10 | 79.8 |
| gpt4o | 3 | 11.1 |

### 4.6 Use of External Tool

While it is certainly possible to solve these problems using a calculator Schick et al. (2023)—and, in fact, the web version of ChatGPT often does so, Arithmetic-Bench is fundamentally different. It is designed to use arithmetic as a proxy for abstract reasoning, providing a controlled setting to evaluate a model's ability to perform multi-step reasoning rather than relying on external tools.

On the other hand, the results of Arithmetic-Bench can be interpreted in two possible ways:

1. In principle, the model's probabilistic predictions are capable of stable multi-step reasoning, but current models have not realized this ability.

2. The model's probabilistic predictions cannot guarantee stable multi-step reasoning. If this is the case, it indicates that using external tools for verification is necessary.

### 4.7 Reproducibility

The results from two independently randomly generated sets of problems show little difference, with average fluctuations below 1%. We will make our code publicly available to ensure reproducibility.

### 5 Conclusion

Arithmetic-Bench provides a rigorous, dynamic, and scalable evaluation of LLMs' multi-step reasoning abilities. Our theoretical analysis shows that the inability to generalize in arithmetic implies broader limitations in general reasoning. Empirical results demonstrate that even state-of-the-art models still struggle with large-number multiplication, highlighting the necessity of improving reasoning mechanisms. We believe that arithmetic-based tasks form the foundation for advancing reasoning in LLMs. Future work should focus on improving data representation, training strategies, verifying, and memory mechanisms. Only by addressing these limitations can LLMs truly perform multi-step reasoning tasks and move beyond shallow inference based on pattern matching.

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

# A  APPENDIX

## A.1  IMPLEMENTATION DETAILS

We conducted experiments using 1–2 machines equipped with 8×A100 GPUs and deployed our models based on vLLM. The number of GPUs required varies depending on the model size. To improve efficiency, we employ parallel inference acceleration for smaller models. Each model was evaluated using the officially recommended decoding parameters. Ablation studies show that the decoding parameters have little effect on the results.

For reasoning models, context length has a significant impact: if the maximum length is insufficient to generate the complete output, performance will decrease a lot. Therefore, it is ultimately set to 16,384. For non-reasoning models, since their responses are naturally short. A maximum length of 2,048 or 4,096 is sufficient.

## A.2  PROOF

Proof of Theorem 1

**Theorem 1.**  A container with capacity $a$ cannot hold information larger than $a$.

**Proof.** Suppose a container with capacity $a_1$ can hold information of size $a_0$, where $a_1 < a_0$. Then there exists a container with capacity $a_2 < a_1$ that can hold the $a_1$-capacity container. Repeating this operation, we can construct a decreasing sequence of capacities $a_n < \cdots < a_2 < a_1 < a_0$.

Since capacities are non-negative, by the monotone bounded sequence theorem, this sequence must have a limit.

**Case 1:** The limit is 0. Then an empty container could hold information of any size, which is obviously a contradiction.

**Case 2:** The limit is greater than 0. Then for each capacity $a$, there exists a corresponding lower bound $b < a$ such that a container of capacity $b$ can hold the container of capacity $a$. Similarly, for $b$, there exists a lower bound $c < b$ such that $c$ can hold $b$, and thus $c$ can hold $a$. This contradicts the assumption that $b$ is the lower bound of the sequence of capacities.

Therefore, the proposition is proved.

## A.3  SUB TASKS

In addition to standard arithmetic operations, we also evaluated several sub-tasks as a complement to the main tasks. As shown in Table 6. The positive correlation between subtask performance and the main task indicates that proficiency on subtasks reflects or contributes to overall performance on the main task.

Most models struggle with sub-tasks such as reversing and counting. Small models (0.5B and 1.5B) exhibit clear flaws in instruction-following, showing low accuracy on tasks like **copy, box, and space**.

## A.4  EXPERIMENTS OF MEMORIZATION

We constructed a dataset using the first 10,000 digits of $\pi$, where the input is the index (the $n$-th digit) and the output is the corresponding digit represented as a one-hot vector. Models with varying parameter sizes were trained to memorize the $\pi$ data up to their capacity limit, defined as the point where the accuracy converges. The learning rate was optimized via grid search to maximize the converged accuracy. We conducted dozens of experiments, and the result of one representative run is shown in Figure 3. The curves from the other experiments exhibit similar shapes.

For different model, we calculated the ratio of the information content of the correctly memorized digits to the total number of model parameters. Our experiments show that this ratio is quite stable, regardless of model size or the number of digits, and is approximately 2.2 bits per parameter.

Table 6: Model performance on Arithmetic-Bench (Sub tasks)

| Model | copy | rev | box | space | length | count | b2d | d2b |
|---|---|---|---|---|---|---|---|---|
| Llama-3-8B-Instruct | 100.0% | 3.7% | 99.8% | 30.2% | 8.6% | 14.4% | 5.5% | 1.9% |
| Llama-3-70B-Instruct | 100.0% | 7.1% | 99.9% | 80.7% | 11.7% | 6.5% | 10.1% | 3.1% |
| Qwen2.5-0.5B-Instruct | 69.2% | 2.8% | 93.1% | 20.3% | 0.2% | 7.8% | 5.0% | 1.3% |
| Qwen2.5-1.5B-Instruct | 69.1% | 9.3% | 99.9% | 82.5% | 6.7% | 6.2% | 2.1% | 3.1% |
| Qwen2.5-3B-Instruct | 99.9% | 13.4% | 99.5% | 86.5% | 9.6% | 4.3% | 8.1% | 2.2% |
| Qwen2.5-7B-Instruct | 99.9% | 13.4% | 99.9% | 99.9% | 13.0% | 24.7% | 12.7% | 3.5% |
| Qwen2.5-14B-Instruct | 99.9% | 17.1% | 100.0% | 100.0% | 15.1% | 24.7% | 12.7% | 3.5% |
| Qwen2.5-32B-Instruct | 100.0% | 23.9% | 100.0% | 99.9% | 16.6% | 42.9% | 15.4% | 4.6% |
| Qwen2.5-72B-Instruct | 99.9% | 14.2% | 99.7% | 100.0% | 23.3% | 32.6% | 15.6% | 5.0% |
| DeepSeek-R1-Distill-Llama-8B | 99.0% | 6.6% | 94.0% | 37.0% | 31.0% | 47.0% | 9.0% | 1.0% |
| DeepSeek-R1-Distill-Llama-70B | 100.0% | 9.0% | 100.0% | 74.0% | 33.0% | 35.0% | 10.0% | 3.0% |
| DeepSeek-R1-Distill-Qwen-1.5B | 96.0% | 10.0% | 77.0% | 17.0% | 34.0% | 13.0% | 3.0% | 2.0% |
| DeepSeek-R1-Distill-Qwen-7B | 100.0% | 14.0% | 91.0% | 93.0% | 36.0% | 44.0% | 20.0% | 5.0% |
| DeepSeek-R1-Distill-Qwen-14B | 100.0% | 24.0% | 100.0% | 100.0% | 38.0% | 59.0% | 11.0% | 2.0% |
| DeepSeek-R1-Distill-Qwen-32B | 100.0% | 23.0% | 100.0% | 100.0% | 25.0% | 42.0% | 13.0% | 4.0% |
| QwQ-32B | 100.0% | 70.0% | 99.0% | 100.0% | 98.0% | 99.0% | 31.0% | 14.0% |
| Qwen3-235B-A22B | 100.0% | 78.0% | 100.0% | 100.0% | 100.0% | 100.0% | 59.0% | 20.2% |
| deepseek r1 671b | 100.0% | 82.0% | 100.0% | 100.0% | 96.0% | 100.0% | 55.0% | 16.0% |
| gpt4 | 100.0% | 15.0% | 100.0% | 100.0% | 54.0% | 22.0% | 11.0% | 3.0% |
| gpt4o | 100.0% | 27.0% | 100.0% | 100.0% | 68.0% | 11.0% | 11.0% | 4.0% |
| gpt3.5 | 100.0% | 20.0% | 90.0% | 51.0% | 17.0% | 10.0% | 3.0% | 2.0% |

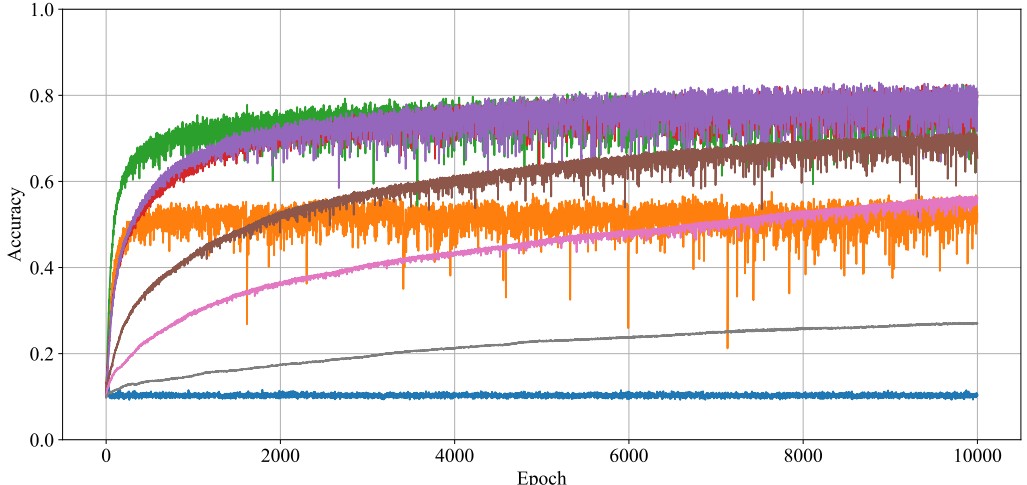

Figure 3: Memorization of $\pi$, different colors represent different learning rates.

### A.5 EXPERIMENTS OF FORGETTING

We constructed datasets using the first 10,000 digits of $\pi$ and $e$. The model was first trained to memorize the $\pi$ dataset up to its capacity limit, and then trained on the $e$ dataset. In the input vectors, the indices of $\pi$ digits were placed on the left, while the indices of $e$ digits were placed on the right, ensuring that the inputs did not overlap.

Despite the absence of input conflicts, the model completely forgot the $\pi$ data after learning $e$. This demonstrates that once a model reaches its capacity limit, adding new information inevitably causes it to forget previously memorized information.

### A.6 USE OF AI ASSISTANTS

To reduce the cost of manual revisions, we used ChatGPT Ouyang et al. (2022) to revise the language of the paper. The revisions were made solely to enhance the clarity and readability of the text and not for any other purpose.

