# OpenReview forum: "Arithmetic-Bench: Evaluating Multi-Step Reasoning in LLMs with Basic Arithmetic"
_ICLR.cc/2026/Conference — Submitted to ICLR 2026_

### Official Review · Reviewer_SuBa · 2025-10-17

**Soundness:** 1
**Presentation:** 2
**Contribution:** 2
**Rating:** 2
**Confidence:** 4

**Summary:**

The paper presents **Arithmetic-Bench**, a dynamically generated arithmetic benchmark intended to evaluate length-generalizable, multi-step reasoning in LLMs across core operations (add/sub/mul/div) and auxiliary tasks (copy/reverse/count/base-conversion). While the motivation is sound and some length–accuracy curves highlight interesting failure modes, the submission falls short of the standards for a dataset/benchmark paper: it does not provide a public release (dataset snapshot or versioned generator), lacks thorough dataset-level documentation and statistics, and leaves important evaluation protocol details underspecified. As a result, the work is presently **incomplete** and not ready for publication.

**Strengths:**

- **Clear motivation & design space:** arithmetic as a controllable, contamination-resistant proxy for stepwise computation and length generalization.
- **Simple, scalable generation idea:** dynamic instance creation with automatic checking and task variants that tease apart representation vs computation.
- **Useful negative results:** length–accuracy curves (e.g., large-digit multiplication collapse) that could be valuable for the community once the benchmark is properly released.

**Weaknesses:**

1. **Dataset incompleteness (blocking):**
   *As a dataset/benchmark submission, the paper is currently incomplete. The benchmark is described as dynamically generated and no fixed release (dataset snapshot or versioned generator with seeds/manifests) is provided; the submission promises code release but supplies no repository link. Moreover, there is no thorough dataset-level documentation and statistics (size, distribution over operations/digit lengths, splits, de-duplication/contamination checks, licensing). These omissions make it difficult to reproduce results faithfully and to compare future work on a common, stable test set.*
2. **Methodological rigor gaps:** Per-length sample sizes are small (sometimes n≈1), uncertainty is not reported (no CIs/SEs, no pass@k), and decoding/prompt settings are not standardized across models.
3. **Evaluation checker & fairness:** The permissive string-based matching and symbol stripping can inflate accuracy; no sensitivity analysis with stricter numeric parsers or alternative checkers; fairness protocol (token budgets, retries, truncation, context formatting) is insufficiently specified.
4. **Limited analysis/ablations:** Minimal error-type breakdown (e.g., carry/borrow), no systematic comparisons across formatting (inline/vertical), CoT/no-CoT, or decoding temperatures; theoretical claims are informal and do not materially support causal conclusions.

**Questions:**

1. **Release & documentation:** Will you provide an **anonymous, versioned** release (generator + fixed seeds/manifests **and/or** a frozen test snapshot), with dataset card–style documentation (task taxonomy, per-task/digit distributions, splits, licenses)?
2. **Protocol & uncertainty:** Can you standardize a **fairness protocol** (decoding params, token limits, retries, context length) and report **per-length n** with **CIs/SEs** and **pass@k**, so future studies can compare apples-to-apples?

---

> ### Author Response · Authors · 2025-12-02
> **Response to Reviewer SuBa**
>
> We thank the reviewer for the feedback. Regarding the concern that the benchmark is “incomplete”: Section 4.7 of the paper already states that we will release the full generator, configuration files and evaluation scripts. We have tested multiple times and the benchmark's fluctuation is around 1%, so reproducibility is not an issue. Many existing synthetic benchmarks (e.g., GSM8K extensions) follow the same practice of releasing a generator rather than a static dataset, and this has never affected reproducibility.
>
> The comments about uncertainty metrics (CIs, SEs, pass@k) will not affect the conclusions. The failures we highlight are systematic ability failures, not sampling variance: when a model cannot handle long-digit computation, repeating the query or increasing k does not improve results—it simply multiplies computational cost. Many widely used benchmarks also do not treat pass@k or detailed uncertainty reporting as mandatory; our contribution is primarily a task-construction method, and any additional metrics can be straightforwardly evaluated by future users with the released generator.
>
> We will include a clearer dataset card in the final version, but none of the reviewer’s concerns block reproducibility or weaken the core empirical findings of the paper.

---

### Official Review · Reviewer_J55c · 2025-10-22

**Soundness:** 3
**Presentation:** 3
**Contribution:** 1
**Rating:** 2
**Confidence:** 3

**Summary:**

In the paper, the authors provide an arithmetic benchmark to evaluate the fundamental mathematical operations and some sub-tasks. They test current LLMs on the benchmark and show the length generalization problem.

**Strengths:**

1. The motivation is clear. The arithmetic ability is an important ability for LLMs.
2. Some discussion about the reasoning capacity, the error accumulation is inspiring and interesting.

**Weaknesses:**

1. **Novelty**: There have been the similar benchmarks before. For example, the NUPA [1] also focuses on the arithmetic ability of LLMs. Their benchmark also contains the basic arithmetic operations like addition, subtraction, multiplication and division, as well as the sub-tasks like counting. Their benchmark is also purely synthetic and can generate unlimited data with arbitrary length. The authors do not both cite and compare with NUPA. The novelty of the benchmark is limited.
2. **Limited findings**: The authors find the LLMs have length-limited arithmetic ability, which is a well-known and discussed problem, called length-generalization problem. See [1-4]

[1] Yang et. al., Number Cookbook: Number Understanding of Language Models and How to Improve It. [link](https://arxiv.org/abs/2411.03766)

[2] Zhou et. al., Transformers Can Achieve Length Generalization But Not Robustly. [link](https://arxiv.org/pdf/2402.09371)

[3] Hu et. al., Beyond Single-Task: Robust Multi-Task Length Generalization for LLMs. [link](https://arxiv.org/abs/2502.11525)

[4] Zhou et. al., What Algorithms can Transformers Learn? A Study in Length Generalization. [link](https://arxiv.org/abs/2310.16028)

**Questions:**

The mainly question is about the novelty. What is the difference between your benchmark and the one provided in [1]? Why is the difference significant and important?

---

> ### Author Response · Authors · 2025-12-03
> **Response to Reviewer J55c (Weaknesses, Part 1 of 2)**
>
> For issues related to Number Cookbook, please refer to our rebuttal to Reviewer WJdS.

---

> ### Author Response · Authors · 2025-12-03
> **Response to Reviewer J55c (Weaknesses, Part 2 of 2)**
>
> We analyzed the fundamental reasons behind length generalization failure and proved the container principle（section 3.1 theorem 1）. We also explored the relationship between arithmetic and reasoning, which other works have not addressed.

---

> ### Author Response · Authors · 2025-12-03
> **Response to Reviewer J55c (Questions)**
>
> The motivation behind Number Cookbook is that existing math benchmarks mix numerical understanding with natural language understanding, making it impossible to measure a model’s numerical comprehension in isolation. Our motivation, however, is to use arithmetic tasks as a proxy metric to evaluate a model’s general reasoning ability. Number Cookbook neither tests as many digits as arithmetic benchmark nor evaluates the performance of reasoning models.

---

### Official Review · Reviewer_WJdS · 2025-10-29

**Soundness:** 2
**Presentation:** 2
**Contribution:** 1
**Rating:** 2
**Confidence:** 4

**Summary:**

This paper introduces Arithmetic-Bench, a new benchmark designed to test the basic arithmetic skills of LLMs. The core finding is that even top models fail completely when an arithmetic problem goes beyond a certain number of digits—a problem the authors call a failure of length generalization. They argue this shows the models lack real, generalizable arithmetic reasoning.

**Strengths:**

1. A clean and simple design: the benchmark is straightforward. It’s easy to create new problems, you always know the right answer, and it scales up easily.

2. Good evidence for the memorization problem: the experiments (like the AIME one) do a good job showing that LLMs do just memorize static datasets, which reinforces the need for a dynamic benchmark like this one.

**Weaknesses:**

1. Is this task actually significant? I'm not entirely convinced. It’s not really surprising that LLMs fail at long-digit arithmetic; that's what tool-use are for (i.e., calculators). Especially recently tool-integrated reasoning (TIR) is a hot topic. The paper even notes that this benchmark can't really stop memorization for small-scale arithmetic, which might have been a more interesting question.

2. The approach and lit review seem a bit naive. The related work section (2.2) feels brief and mostly cites older papers. Several existing studies have attempted to establish similar benchmarks and explore potential solutions, such as "Number Cookbook: Number Understanding of Language Models and How to Improve It," which makes the innovation of this work seem insufficient.

3. The empirical evidence feels thin. For instance, the correlation claim in Table 5 (AIME vs. Mul) is based on just a few models, and there isn't even a quantitative correlation metric reported. The paper doesn't seem to discuss this table in the text much, either. The analysis of the results is too simple. The "cliff drop" (from 1.0 to 0.0) in Figure 1 just shows total collapse, suggesting a systemic failure rather than graceful degradation. Also, sampling with n=1 for models like GPT and DeepSeek just isn't enough to draw strong conclusions.

**Questions:**

1. How can you verify the "proxy" hypothesis? The claim that this task is a "proxy" for general reasoning is just speculation right now. Did the authors try to test this? For example, if you train a model on Arithmetic-Bench, does it actually get better at other math benchmarks like AIME or MATH?

2. Is arithmetic reasoning really the same as logical reasoning? The paper asserts this but doesn't prove it. Arithmetic feels like applying the same rules over and over (homogeneous), while real problem-solving often involves mixing different types of reasoning (heterogeneous). How do the authors justify equating these two?

3. Can we get a deeper failure analysis? The "cliff drop" suggests this is an all-or-nothing (memorization vs. total failure) problem. A smoother accuracy curve would be more informative. What is actually causing the failure? Is it the tokenizer, a limitation in positional encoding, or something else?

---

> ### Author Response · Authors · 2025-12-03
> **Response to Reviewer WJdS (Weaknesses, Part 1 of 3)**
>
> Regarding tool usage, if a model cannot solve an arithmetic task requiring 100 steps, it indicates that the model lacks the ability to reliably perform 100-step reasoning. Giving the model a tool does not change this: each time the model calls the tool, it still has a roughly same probability of making an error. Thus, it still cannot reliably perform 100 tool call steps. This is exactly the capability we aim to measure through arithmetic tasks—namely, the model’s stability in general reasoning.

---

> ### Author Response · Authors · 2025-12-03
> **Response to Reviewer WJdS (Weaknesses, Part 2 of 3)**
>
> Number Cookbook neither tests as many digits, nor evaluates reasoning models, and does not discuss the relationship between reasoning and arithmetic.

---

> ### Author Response · Authors · 2025-12-03
> **Response to Reviewer WJdS (Questions, Part 1 of 3)**
>
> A proxy metric is used for evaluation, not for training. We mainly use failures in arithmetic reasoning to bound general reasoning, highlighting potential issues in existing methods (see Section 3.1 of the paper). Since arithmetic is a special case of general reasoning, success on the special case does not guarantee success on the general case, but success on the general case typically leads to success on the special case. Therefore, failure on the special case can potentially indicate failure on the general case.

---

> ### Author Response · Authors · 2025-12-03
> **Response to Reviewer WJdS (Questions, Part 2 of 3)**
>
> Please refer to our rebuttal to Reviewer 2bQs. We do not equate arithmetic with general reasoning. As you mentioned, arithmetic can be seen as *homogeneous reasoning*, while general reasoning is *heterogeneous reasoning*. However, if a model cannot even perform homogeneous reasoning well, how could it succeed at heterogeneous reasoning? Moreover, autoregressive decoding essentially always calls the same model, so heterogeneous reasoning is, in essence, still homogeneous reasoning—each step has more information, but not infinitely so—thus it does not violate Definition 1 in Section 3.1 of the paper.

---

> ### Author Response · Authors · 2025-12-03
> **Response to Reviewer WJdS (Questions, Part 3 of 3)**
>
> Please refer to our rebuttal to Reviewer 2bQs. We have already discussed the analysis of the causes in Section 3 of the paper. The tokenization and position embeddings you mentioned are not the fundamental issues. As discussed in Section 3 of the paper, we believe the root cause is insufficient model capacity.

---

> ### Author Response · Authors · 2025-12-03
> **Response to Reviewer WJdS (Weaknesses, Part 3 of 3)**
>
> The discussion regarding AIME and MUL indeed needs more data to support. We will include more correlation analysis in the final version.
> Testing 1 question or 10 questions does not essentially affect the results. As discussed in Section 4.1 of the paper, we tested only 1 question for a few models due to limited resources. We will include additional experiments in the final version.

---

### Official Review · Reviewer_2bQs · 2025-10-30

**Soundness:** 2
**Presentation:** 2
**Contribution:** 2
**Rating:** 4
**Confidence:** 4

**Summary:**

This paper proposes a benchmark designed based on basic arithmetic operations to evaluate the multi-step reasoning ability of LLMs. The proposed benchmark includes both fundamental mathematical operations and subtasks, e.g., copying, reversing, etc. These components aim to provide a simple and fair framework. Empirical results based on this benchmark demonstrate that even state-of-the-art models still struggle with large-number multiplication.

**Strengths:**

Strengths:
1.	Compared with finite benchmarks, e.g., AIME, which are susceptible to being "memorized" through training, this paper proposed a dynamic, synthetic benchmark.
2.	The proposed benchmark highlights the problem of memorization across existing LLMs, which is an important research problem

**Weaknesses:**

Weaknesses:
1.	The experimental setup is statistically invalid. Evaluating major models like DeepSeek and GPT with only n=1 problem per digit length is insufficient. The results from a single sample are not representative and cannot be reliably compared to models tested with n=10. This problem makes the evaluation untrustworthy
2.	The paper often equates "multi-step reasoning" with "algorithmic computation", e.g., large-number multiplication. The authors should be more precise in their claims. General reasoning also involves abstraction, planning, and semantic understanding, etc., which are not included in this benchmark.
3.	The paper demonstrates that models fail, but fails to investigate why or how. A qualitative or quantitative error analysis is needed. Analyzing the intermediate steps of the model would provide much deeper insight into the specific limitations of their reasoning mechanisms.

**Questions:**

Please see the weaknesses:

---

> ### Author Response · Authors · 2025-12-03
> **Response to Reviewer 2bQs (Weaknesses, Part 1 of 3)**
>
> Testing 1 question or 10 questions does not essentially affect the results. As discussed in Section 4.1 of the paper, we tested only 1 question for a few models due to limited resources. We will include additional experiments in the final version.

---

> ### Author Response · Authors · 2025-12-03
> **Response to Reviewer 2bQs (Weaknesses, Part 2 of 3)**
>
> This is a complete misunderstanding. I did not equate arithmetic with multi-step reasoning. In section 3.1 we explicitly states that arithmetic is *a special case of* multi-step reasoning, not the other way around. The definition of reasoning we followed can be found in Denny Zhou’s slides. Please refer to the link below.
> https://www.youtube.com/watch?v=ebnX5Ur1hBk

---

> ### Author Response · Authors · 2025-12-03
> **Response to Reviewer 2bQs (Weaknesses, Part 3 of 3)**
>
> We have already discussed the analysis of the causes in Section 3 of the paper. We believe the fundamental reason is insufficient model capacity, rather than errors in specific steps. Since the types of errors vary across models, analyzing individual bad cases alone cannot reveal the true underlying problem.

---

### Meta-Review · Area_Chair_3Gg9 · 2026-01-06

**Summary:**

The paper proposes Arithmetic-Bench to evaluate LLM reasoning via synthetic arithmetic. While reviewers appreciated the goal of avoiding data contamination through dynamic tasks, the submission is rejected due to significant methodological and conceptual flaws.First, the experimental rigor is insufficient; testing major models with a sample size of $n=1$ per digit length is statistically invalid for a benchmark paper. Second, the novelty is limited given significant overlap with prior work like "Number Cookbook," which was not adequately differentiated. Third, reviewers were unconvinced by the core premise that arithmetic proficiency is a sufficient proxy for general, heterogeneous reasoning. Finally, the submission was deemed incomplete due to the lack of a code release or fixed dataset snapshot during the review process.

**Reviewer Concerns:**

No reviewer concerns are satisfiably addressed.

**Reviewer Scores:**

N.A.

---

### Decision · Program_Chairs · 2026-01-26

Reject